# LEARNING COMPACT VISION TOKENS FOR LARGE MULTIMODAL MODELS

## ABSTRACT

Large multimodal models (LMMs) suffer significant computational challenges due to the high cost of Large Language Models (LLMs) and the quadratic complexity of processing long vision token sequences. In this paper, we explore the spatial redundancy among vision tokens and shorten the length of vision token sequences for inference acceleration. Specifically, we propose a Spatial Token Fusion (STF) method to learn compact vision tokens for short vision token sequence, where spatial-adjacent tokens are fused into one. Meanwhile, weight-frozen vision encoder can not well adapt to the demand of extensive downstream vision-language tasks. To this end, we further introduce a Multi-Block Token Fusion (MBTF) module to supplement multi-granularity features for the reduced token sequence. Overall, we combine STF and MBTF module to balance token reduction and information preservation, thereby improving inference efficiency without sacrificing multimodal reasoning capabilities. Experimental results demonstrate that our method based on LLaVA-1.5 achieves comparable or even superior performance to the baseline on 8 popular vision-language benchmarks with only 25% vision tokens of baseline. The source code and trained weights will be publicly available.

## 1 INTRODUCTION

Large multimodal models (LMMs) (Liu et al., 2023; 2024a) based on Large Language Models (LLMs) (Achiam et al., 2023; Team et al., 2023; Touvron et al., 2023) have shown remarkable multimodal reasoning capabilities on various application (Dang et al., 2023). LMMs leverage a vision encoder such as CLIP-ViT (Radford et al., 2021) to embed images into vision tokens as the prefix visual context, and feed them into a large language model pretrained on large-scale text corpus. Despite the impressive capabilities, LMMs face substantial computational challenges, which limit the scalability and efficiency.

The high computation cost of LMMs primarily comes from LLMs, where vision encoder for LMMs is obviously smaller than the corresponding LLM. For example, CLIP-ViT-L adopted LLaVA (Liu et al., 2023; 2024a) only has 0.3B parameters, while the corresponding LLM, such as LLaMA (Touvron et al., 2023) or Vicuna (Chiang et al., 2023), contains 7B or 13B parameters. Although using LLMs with fewer parameters, such as Phi-2 (Javaheripi et al., 2023), can alleviate this burden, it often leads to obvious performance drops on visual question-answering and reasoning.

Another solution to improve inference efficiency of LMM is to reduce the number of vision tokens fed into LLM. Since the number of vision tokens produced by vision encoder reaches hundreds even thousands, significantly surpassing the number of text tokens, the reduction of vision token can significantly improve the inference efficiency of LMM. As a result, several methods (Li et al., 2024a;b; Shang et al., 2024) are proposed to shorten the length of vision token sequence, which are fed into LLM. However, the balance between the token reduction and information retention remains an open question.

In this paper, we explore the spatial redundancy present in the visual context of LMMs. As shown in Figure 1, we retrain LLaVA-1.5-7B by simply reducing the number of vision tokens using average pooling with a stride of 2 before fed into LLM, where adjacent $2 \times 2$ tokens are averaged as one and only 25% tokens are remained. The reduced model is designated as LLaVA-1.5-7B (AvgPool). We surprisingly find that its performance drop on several popular benchmarks is not obvious, except

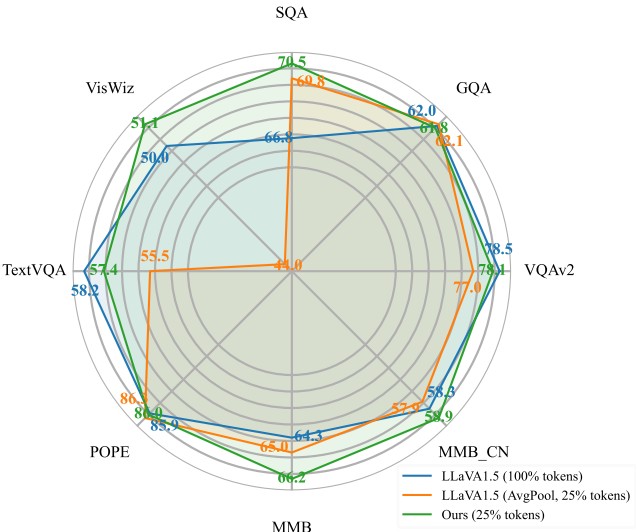

Figure 1: Spatial token redundancy in the visual context of LMMs. The performance gap between LLaVA-1.5-7B (AvgPool) with only 25% vision tokens and the original LLaVA-1.5-7B is not so obvious, except VisWiz, demonstrating the excessive redundancy of vision tokens. Based on STC and MLTC, our method achieves comparable or even better performance than the original LLaVA-1.5-7B.

VisWiz (Gurari et al., 2018). Especially on POPE (Li et al., 2023b) and GQA (Hudson & Manning, 2019) benchmarks, LLaVA-1.5-7B (AvgPool) even outperforms the original LLaVa-1.5-7B. The results support that excessive redundancy indeed exists in current LMMs.

To reduce the redundancy of vision tokens, we propose a *Spatial Token Fusion* (*STF*) method, which fuses adjacent vision tokens into one to shorten the token sequence. Unlike previous methods, we address the spatial redundancy before vision tokens are fed into LLM. Instead of simply averaging adjacent vision tokens, our approach concatenates adjacent $k^2$ vision tokens in the sliding window with size of $k \times k$ along the channel dimension. Then, we introduce learnable *Spatial Token Fusion* (*STF*) module to adaptively fuse features of adjacent $k^2$ vision tokens, while bridging the representations between vision encoder and LLM. Compared to plain average pooling, our approach aims to preserve more information during redundancy reduction.

Since vision encoder of LLaVA-style LMM is generally fixed during training, vision tokens generated by vision encoder can not well adapt to the demand of target tasks, especially for some tasks that require detail information of the give image. To capture more detail visual information of image, we further propose a *Multi-Block Token Fusion* (*MBTF*) module to integrate low-level features with high-level semantic features, thus improving the compactness of the fused vision tokens. In this manner, our method can adaptively access multi-level features from vision encoder for widespread downstream vision-language tasks without the retraining of vision encoder. Moreover, compared to LLM, the computation cost of additional modules, including STF and MBTF, can be ignored, yet the reduced sequence of vision tokens can significantly accelerate the inference of LMM.

In summary, the contributions of our method can be summarized as below.

- We propose a Spatial Token Fusion module, which learns compact vision tokens to significantly shorten the vision token sequence fed into LLMs, thus effectively accelerating the inference of LMMs.

- We propose a Multi-Block Token Fusion module to adapt the feature demand of extensive downstream vision-language tasks without the retraining of vision encoder.

- Extensive experiments on LLaVA (Liu et al., 2023) show that our approach achieves comparable or even superior performance to LLaVA-1.5-7B (Liu et al., 2024a) on popular vision-language benchmarks with only 25% vision tokens of the original one.

## 2 RELATED WORK

### 2.1 ACCELERATION OF LMMS

Efforts to optimize the efficiency of LMMs (Shao et al., 2025) have explored diverse strategies, such as lightweight vision encoders (Dosovitskiy et al., 2020; Fang et al., 2023; Haotian et al., 2024), sparsely activated MoE architectures (Zhang et al., 2024c), and parameter-efficient language models (Javaheripi et al., 2023). Among these, vision token pruning (Li et al., 2024b; Guo et al., 2024; Yu et al., 2024b) has gained traction due to its ability to shorten visual sequences without altering model parameters. For example, LLaVA-PruMerge (Shang et al., 2024) merges redundant tokens at CLIP's penultimate layer, while FastV (Chen et al., 2024b) employs adaptive attention patterns to prioritize essential tokens and prune others. Concurrently, LLaVolta (Chen et al., 2024a) introduces progressive token compression across training stages, balancing efficiency and performance. TokenPacker (Li et al., 2024a) proposes a coarse-to-fine visual projector that hierarchically compresses high-resolution image features through downsampling, point-region interaction, and cross-layer fusion to generate compact vision tokens. TinyChart-3B's vision token merging (Zhang et al., 2024a) dynamically reduces high-resolution input processing overhead by fusing similar vision tokens within each transformer layer. YOPO (Zhang et al., 2024d) integrates three strategies, including 126 neighbor-aware vision token attention, pruning of inactive visual attention heads, and selective layer dropping for visual computations, to improve the inference efficiency of LMMs. VoCo (Ye et al., 2025) leverages LLM attention distillation to conduct vision token compression. VISTA (Li et al., 2025) introduces visual information steering with token-logit augmentation strategy to reduce hallucination of LMMs. Yu et al. (2024a) proposes a dynamic pruning strategy to identify inflection point in vision class token similarity curve for vision token reduction. SEED (Ge et al.) introduces an elaborate image tokenizer to empower LMM to achieve image understanding and image generation. CrossGET (Shi et al., 2023) adopts cross-modal guided token matching and ensemble algorithm to adaptively combine tokens for inference acceleration. Mini-Gemini (Li et al., 2024c) follow the idea of $S^2$ (Shi et al., 2024) to enhance the performance of LMMs without increment of vision tokens.

VisionZip (Yang et al., 2025a) selects a set of informative tokens to reduce vision token redundancy. SparseVLM (Zhang et al., 2024b) proposes text-guided training-free token reduction strategy to accelerate VLM inference. MMTok (Dong et al., 2025) selects informative vision tokens by solving a maximum coverage problem, thus reducing vision tokens. LFTR (Zhao et al., 2025) introduces a learning-free token reduction to accelerate LMM inference without additional finetuning. MustDrop (Liu et al., 2024b) measures the importance of vision token from the whole inference lifecycle, thus achieving more accurate vision token reduction. Elastictok (Yan et al., 2024) adaptively encodes video or image into variable number of tokens for efficient multimodal inference.

In spite of encouraging performance achieved, the above methods require the involvement of text tokens to prune the number of vision tokens, thus compromising model performance due to the loss of fine-grained visual details. In comparison, our method fuses adjacent vision tokens for information preservation, while adaptively accessing to features from different layers for widespread vision-language tasks.

### 2.2 ACCELERATION OF LLMS

The quadratic computational complexity of transformer-based LLMs, which scales with the square of the input sequence length (Vaswani et al., 2017), has motivated substantial efforts to address inherent redundancy in these architectures. Prior research has explored two primary directions: parameter sparsification through weight pruning (Goyal et al., 2017a; Frantar & Alistarh, 2023) and attention head reduction (Michel et al., 2019), and sequence compression to mitigate the overhead of long token sequences. For the latter, hierarchical approaches like Pyramid Transformers (Dai et al., 2020) progressively downsample token sequences across layers, while Nawrot et al. (Nawrot et al., 2022) propose adaptive sequence compression by semantic boundary prediction. Recent VCC (Zeng et al., 2023) further introduces layer-wise token aggregation by select important tokens.

However, the integration of visual encoders with LLM decoders introduces modality-specific computational bottlenecks, particularly in processing lengthy vision token sequences derived from high-resolution images. Unlike unimodal compression that prioritizes linguistic patterns, vision-language

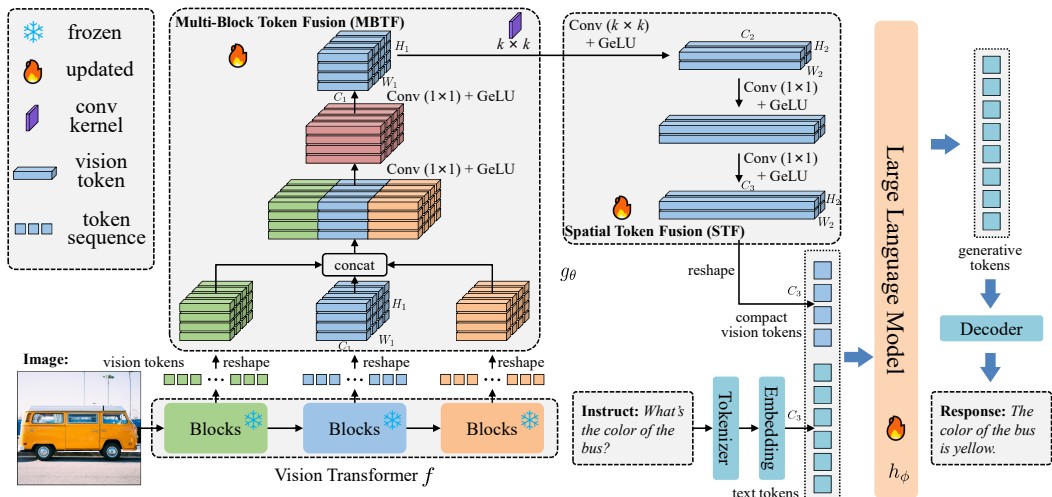

Figure 2: The overview of our method. Our method follows LLaVA-style architecture and introduces additional Multi-Block Token Fusion (MBTF) and Spatial Token Fusion (STF) modules. Based on basic LLaVA, we obtain vision tokens produced by selected intermediate blocks of vision encoder and then fuse them along the channel dimension for multi-granularity features. Then, the fused vision tokens are fed into Spatial Token Fusion module, where convolution with kernel size of $k \times k$ is applied on the above vision tokens to aggregate vision tokens in $k \times k$ neighborhood and obtain more compact vision tokens, thus reducing spatial redundancy of input vision token sequence for large language model. Finally, both compact vision tokens and text tokens from input instruct are fed into large language model to generate the corresponding response.

interactions demand modality-aware token reduction to preserve critical spatial-semantic correlations. To bridge this gap, we propose a novel vision token compression framework that strategically reduces the number of vision tokens fed into the LLM component of LMMs.

## 3 METHOD

In this section, we first introduce the preliminaries of LLaVA-style large multimodal model. Then, we give a brief overview of our method. Afterward, we present our proposed Multi-Block Token Fusion module and Spatial Token Fusion in detail. Finally, we depict the optimization of our method.

### 3.1 PRELIMINARIES

Given an input image $X_v$, the vision tokens $X_v^l = f_l(X_v)$ is the output of the $l$-th block of the vision encoder $f$. To bridge the gap between image and language modalities, LLaVA introduces a linear- or MLP-based projector $g_\theta$ ($\theta$ is the parameters of projector) to map the vision tokens $X_v^l$ into the text embedding space and obtains aligned tokens $Z_v = g_\theta(X_v^l)$, which have the same dimension as the text embedding in large language model $h_\phi$. Then, aligned tokens $Z_v$ and instruct $X_{\text{instruct}}$ are fed into the LLM $h_\phi$ to generate response as $X_{\text{response}} = h_\phi(Z_v, X_{\text{instruct}})$.

In this paper, we aim to reduce the number of aligned tokens $Z_v$ in a lossless manner to shorten the length of vision token sequence fed into LLM, thus accelerating the inference of LMM. To this end, we focus on the design of projector $g_\theta$ in the following sections.

### 3.2 OVERVIEW

The goal of token reduction is to minimize the length of vision token sequence, while maximizing the performance of the reduced large multimodal model. To accomplish this goal on LMM, we explore the solution to reduce spatial redundancy among vision tokens, which stems from the local similarity property of natural images, especially for high-resolution images. Meanwhile, we also consider information preservation during redundancy reduction, thus minimizing the performance drop.

As shown in Figure 2, we adopt a dual stage token fusion strategy to implement the projector $g_\theta$ between vision encoder and LLM. In the first stage, we extract vision tokens from the selected blocks of vision encoder. The extracted multi-block vision tokens are fused into compact one, which contains semantic representations with various granularities, thus preserving information for extensive downstream tasks. In the second stage, we further fuse the above compact tokens in the neighborhood to reduce the spatial redundancy. After the above two token fusion steps, the fused vision tokens are further aligned to text embedding space of large language model. Finally, combined with text tokens of instruct, the fused vision tokens are fed into LLM to generate the corresponding response in a significant efficient manner.

### 3.3 MULTI-BLOCK TOKEN FUSION

In this section, we focus on the integration of multi-block tokens from vision encoder, to obtain multi-granularity representations. Our main target is to learn compact vision tokens, which embrace multi-level semantic features of vision encoder, thereby adaptive to extensive vision-language tasks.

To fuse representative features from vision encoder $f$, we select vision tokens from $M$ blocks, whose indices can be denoted as $\{l_i\}_{i=1}^M$. Since the features from adjacent blocks have similar semantics, directly fusing of all these features would introduce substantial computation cost. Hence, the indices $\{l_i\}_{i=1}^M$ of selected blocks are evenly sampled. For LLaVA-1.5-7B, its vision encoder, ViT-L/14, has 24 blocks, and the indices of sampled blocks are $\{3, 6, 9, 12, 15, 18, 21, 24\}$.

As depicted in Figure 2, we introduce Multi-Block Token Fusion (MBTF) module to fuse multi-block tokens as follows. First, we reshape the vision tokens, $\{X_v^{l_i}\}_{i=1}^M$, as feature maps for better understanding of follow-up operation. Then, the selected features are concatenated along channel dimension. Next, we fuse the concatenated features using two sequential convolution modules with kernel size of $1 \times 1$, followed by GeLU (Hendrycks & Gimpel, 2016) activation function. To improve the compactness of the fused vision tokens, we progressively reduce the size of channel dimension in the convolution step. For LLaVA-1.5-7B, the channel dimension size of the above convolution operations are 4096 and 1024, respectively, where the token dimension of vision encoder is 1024. The overall process of MBTF can be presented as

$$X_v^{\text{MBTF}} = \text{conv}\left(\text{conv}\left(\text{concat}\left(\{X_v^{l_i}\}_{i=1}^M\right)\right)\right), \tag{1}$$

where $\text{conv}$ denotes the convolution with kernel size of $1 \times 1$, followed by GeLU.

### 3.4 SPATIAL TOKEN FUSION

In this section, we further reduce spatial redundancy of the above multi-block fused vision tokens.

Let $H_1 \times W_1 \times C_1$ denotes the shape of multi-block fused vision tokens $X_v^{\text{MBTF}}$, where $H_1$, $W_1$ and $C_1$ are the height, width and channel size of $X_v^{\text{MBTF}}$, respectively. The text embedding of LLM is set to $C_3$. Generally, the token dimension of LLM adopted in LMM is significantly larger than the one of the corresponding vision encoder, namely $C_3 \gg C_1$. An intuitive idea to shorten the length of vision token sequence is concatenating multiple vision tokens as one, whose dimension is close to the one of text token, thereby fusing multiple vision tokens without loss of information.

As depicted in Figure 2, we introduce Spatial Token Fusion module to reduce the redundancy of vision tokens. Specifically, we adopt convolution operation with kernel size of $k \times k$ to fuse the $k^2$ tokens of $X_v^{\text{MBTF}}$ into a compact token as follows

$$X_1^{\text{STF}} = \text{conv}_{k \times k}\left(X_v^{\text{MBTF}}\right), \tag{2}$$

where $\text{conv}_{k \times k}$ denotes the convolution operation with kernel size of $k \times k$ and stride size of $k \times k$, followed by GeLU. The size of $X_1^{\text{STF}}$ is $H_2 \times W_2 \times C_2$, where $H_2 = \frac{H_1}{k}$ and $W_2 = \frac{W_1}{k}$. If $\text{conv}_{k \times k}$ is regarded as a learnable concatenation operation, we set $C_2 = k^2 \cdot C_1$. For LLaVA-1.5-7B, the dimension of vision token and text token are 1024 and 4096, respectively. If we set $k = 2$, the dimension of fused vision token $C_2$ is identical to the one of text token $C_3$ ($C_2 = C_3 = 4096$), thus achieving lossless token reduction.

For more general solution, adjacent $k^2$ vision tokens can be also fused to any number of tokens, by tensor reshaping. Specifically, the fused token $X_1^{\text{STF}} \in \mathbb{R}^{H_2 \times W_2 \times C_2}$ can be reshaped as $E$ tokens,

where the shape of each token is $H_2 \times W_2 \times \frac{C_2}{E}$. This provides more flexibility for spatial token fusion.

To align the fused vision tokens with text embedding of LLM, we further introduce two additional convolution modules with kernel size of $1 \times 1$, followed by GeLU activation function and obtain aligned vision token sequence $X_v^{\mathrm{STF}} \in \mathbb{R}^{(\frac{H_1}{k} \cdot \frac{W_1}{k} \cdot E) \times C_3}$. Finally, the number of aligned vision tokens fed into LLM, is reduced from $(H_1 \cdot W_1)$ to $\left( \frac{H_1}{k} \cdot \frac{W_1}{k} \cdot E \right)$, where adjacent $k^2$ vision tokens are fused into $E$ tokens and $E < k^2$.

## 3.5 Optimization

Following the training scheme of LLaVA (Liu et al., 2023; 2024a), we optimize our reduced LMM model by maximizing the probability $p$ of generating target response $X_r$ by

$$p(X_r|X_v, X_{\mathrm{instruct}}) = \prod_{i=1}^{N} p_{\theta,\phi}(x_i|X_v^{\mathrm{STF}}, X_{\mathrm{instruct}}, X_{r,<i}), \tag{3}$$

where $\theta$ and $\phi$ are learnable parameters of projector $g_\theta$ and LLM $h_\phi$, $N$ is the length of response $X_r$ and $X_{r,<i}$ are response tokens before the current prediction token $x_i$.

We also follow the two-stage optimization scheme of LLaVA, including feature alignment pretraining and end-to-end finetuning. In the pretrainig stage, only parameters $\theta$ are updated and others are fixed. In the finetuning stage, only parameters $\phi$ are learnable and other parameters are frozen.

## 4 Experiments

First, we present our experiment setup, including model architecture, evaluation benchmarks and implementation details. Afterward, we evaluate our models on extensive popular vision-language benchmarks and compare with other efficient LLaVA models. Then, we conduct ablation studies to dissect the key components of our approach. Finally, we give some cases of vision-language reasoning and compare our method and the baseline LLaVA.

### 4.1 Experiment Setup

#### 4.1.1 Model Architecture

In this work, we mainly focus on the acceleration of LLaVA models (Liu et al., 2023; 2024a). Our models use the CLIP ViT-L/14 (Radford et al., 2021) as the vision encoder and the resolution of input image is set to $336 \times 336$. Besides, we utilize Vicuna-1.5-7B (Chiang et al., 2023) as the LLM backbones. The number of selected blocks $M$ for Multi-Block Token Fusion module is set to 8. By default, the kernel size $k$ for Spatial Token Fusion is set to $2 \times 2$ and the number of target fused tokens $E$ is set to 1. To fully integrate information from different layers, we select every three layer from the 24 blocks of the vision encoder. The structures of Multi-Block Token Fusion and Spatial Token Fusion for LLaVA-1.5-7B are listed in Table 1, respectively.

Table 1: The model structures of MBTF and STF. "act" denotes the activation function after the convolution module.

| layer | output size | kernel size | stride | act |
|---|---|---|---|---|
| conv1 | $24 \times 24 \times 4096$ | $1 \times 1$ | 1 | GeLU |
| conv2 | $24 \times 24 \times 1024$ | $1 \times 1$ | 1 | GeLU |

(a) Multi-Block Token Fusion (MBTF)

| layer | output size | kernel size | stride | act |
|---|---|---|---|---|
| conv1 | $12 \times 12 \times 4096$ | $2 \times 2$ | 2 | GeLU |
| conv2 | $12 \times 12 \times 16384$ | $1 \times 1$ | 1 | GeLU |
| conv3 | $12 \times 12 \times 4096$ | $1 \times 1$ | 1 | GeLU |

(b) Spatial Token Fusion (STF)

We compare our method with 6 efficient LLaVA-style methods, including LLaVA-1.5 (Liu et al., 2023), PruMerge+ (Shang et al., 2024), FastV (Chen et al., 2024b), LLaVolta (Chen et al., 2024a), YOPO (Zhang et al., 2024d) and LLaVA-Mini (Zhang et al., 2025). All above methods use CLIP ViT-L/14 as the vision encoder, and Vicuna-1.5-7B as the LLM backbones. For fairness comparison, all methods follow the data preparation of LLaVA-1.5 (Liu et al., 2023).

Table 2: Performance comparison on 8 popular vision-language reasoning benchmarks. $\text{VQA}^T$ is short for TextVQA. "Avg." denotes the average score of the previous 8 vision-language benchmarks. Method marked with "*" is reproduced by us according to the official code. Our method is finetuned on full parameters of LLM, instead of LoRA. The best results are marked by **bold font**. The second-best results are marked by underline.

| Backbone | Method | TFLOPs | GQA | SQA | $\text{VQA}^T$ | POPE | MMB | $\text{MMB}^{CN}$ | $\text{VQA}^{v2}$ | VisWiz | Avg. (%) |
|---|---|---|---|---|---|---|---|---|---|---|---|
| | LLaVA-1.5 (lora) (Liu et al., 2024a) | 7.6 | 63.0 | 68.4 | 58.2 | 86.4 | 66.1 | 58.9 | 79.1 | 47.8 | 66.0 |
| | LLaVA-1.5 (full tuning) (Liu et al., 2024a) | 7.6 | 62.0 | 66.8 | 58.2 | 85.9 | 64.3 | 58.3 | 78.5 | 50.0 | 65.5 |
| | LLaVolta (Chen et al., 2024a) | 5.0 | 62.1 | 70.5 | 58.7 | 86.3 | 65.6 | 59.9 | 78.8 | 48.3 | 66.3 |
| | PruMerge+ (Shang et al., 2024) | 1.9 | 59.3 | 68.3 | 57.1 | 84.0 | 64.9 | 53.2 | 76.8 | 49.8 | 64.2 |
| Vicuna-1.5-7B | FastV (Chen et al., 2024b) | 1.9 | 60.3 | - | **57.7** | 83.2 | 64.3 | 58.0 | 77.7 | 50.8 | 64.6 |
| | YOPO (Zhang et al., 2024d) | 1.9 | 61.6 | 69.0 | 56.3 | **86.8** | 65.5 | **59.6** | 78.0 | 49.9 | 65.8 |
| | LLaVA-Mini* (Zhang et al., 2025) | 1.9 | 58.4 | 67.2 | 55.2 | 83.2 | 63.2 | 55.3 | 76.2 | 48.2 | 63.4 |
| | LLaVA-1.5 (AvgPool) | 1.9 | **62.1** | 69.8 | 55.5 | 86.3 | 65.0 | 57.9 | 77.0 | 44.0 | 64.7 |
| | **STC (ours)** | 1.9 | 61.9 | **70.5** | 57.4 | 86.0 | **66.2** | 58.9 | **78.1** | **51.1** | **66.3** |

### 4.1.2 EVALUATION BENCHMARKS

We evaluate our method on 8 popular vision-language benchmarks, including GQA (Hudson & Manning, 2019), ScienceQA (SQA) (Lu et al., 2022), VQAv2 (Goyal et al., 2017b), VisWiz (Gurari et al., 2018), TextVQA (Singh et al., 2019), POPE (Li et al., 2023b), MMBench (Liu et al., 2024c) and MMBench-CN (Liu et al., 2024c). GQA (Hudson & Manning, 2019) tests fine-grained visual reasoning with multistep question-answer pairs. SQA (Lu et al., 2022) with multiple choice are used to evaluate the zero-shot generalization on scientific question answering, and we focus on the SQAI subset, which contains questions that specifically include images as part of the question context. VQAv2 (Goyal et al., 2017b) tests visual and commonsense reasoning with open-ended questions on images. VizWiz (Gurari et al., 2018) contains 8,000 images to evaluate model's zero-shot generalization on visual questions asked by visually impaired people. TextVQA (Singh et al., 2019) contains text-rich visual question answering. POPE (Li et al., 2023b) measures object hallucination under varying conditions, and we report the average F1 score on all conditions. MMBench (Liu et al., 2024c) and the CN version (Liu et al., 2024c) evaluate a model's answer robustness with all-round shuffling on multiple choice answers.

### 4.1.3 IMPLEMENTATIONS

We follow LLaVA-1.5 (Liu et al., 2023) to perform data preparation and training schedule for pre-training and instruction tuning, and train the model from scratch with reduced spatial visual redundancy. We pretrain our model on the filtered CC-595K (Liu et al., 2023) subset for 1 epoch with learning rate of $1 \times 10^{-3}$ and batch size of 256, and finetune on the proposed LLaVA-Instruct-158K (Liu et al., 2023) dataset for 1 epoch, with learning rate of $2 \times 10^{-5}$ and batch size of 128. The Adam optimizer is employed without weight decay, and the learning rate follows cosine schedule with warmup ratio of 3%. For efficient GPU memory usage during finetuning, we utilize Deep-Speed (Rasley et al., 2020) and gradient checkpointing, without offloading. Additionally, bfloat16 and TensorFloat32 are enabled to strike a balance between computational speed and precision. In the pretraining stage, we update the parameters of both MBTF and STF module, but fix the ones of LLM. In the instruction tuning stage, we update the full parameters of LLM, MBTF and STF. During pretraining and finetuing, the parameters of vision encoder are always frozen. We conduct all above experiments on servers, each of which contains $8 \times$ Nvidia RTX A6000 GPUs.

### 4.2 COMPARISON TO OTHER EFFICIENT LLAVA MODELS

To validate the effectiveness of our method, we apply our method on LLaVA model and compare it with other efficient LLaVA models. As shown in Table 2, our method achieves the best average performance on 8 popular vision-language reasoning benchmarks among methods, which have a similar computation cost about 1.9 TFLOPs (evaluated by calflops (Ye, 2023)). In spite of only 25% vision tokens of the original LLaVA model used, our method even outperforms both LLaVA-1.5 (lora) and LLaVA-1.5 (full tuning), which adopt full sequence of vision tokens, by 0.3% and 0.8% average score, respectively. The performance gain stems from high scores on SQA, MMB, VQAv2 and VisWiz benchmarks, especially on SQA. The experimental results demonstrate that our method can effectively reduce the spatial redundancy of vision tokens and accelerate the inference of LLaVA model, while maintaining comparable, even better performance.

Surprisingly, even simply applying average pooling on vision tokens (reduced to 25% of the original one) and retraining the LLaVA model, it also acheves better performance than several other effecient LLaVA models. It indeed supports excessive spatial redundancy among vision tokens of LLaVA.

Table 3: The effect of fusion modules. All results are evaluated on the baseline of LLaVA-1.5 with Vicuna-1.5-7B. "Avg." denotes the average score of the previous 8 vision-language benchmarks.

| Method | TFLOPs | GQA | SQA | VQA$^T$ | POPE | MMB | MMB$^{CN}$ | VQA$^{v2}$ | VisWiz | Avg. (%) |
|---|---|---|---|---|---|---|---|---|---|---|
| LLaVA-1.5 (baseline, full tuning) (Liu et al., 2024a) | 7.6 | 62.0 | 66.8 | 58.2 | 85.9 | 64.3 | 58.3 | 78.5 | 50.0 | 65.5 |
| MBTF | 7.6 | 63.2 | 69.3 | 58.9 | 86.8 | 65.3 | 59.5 | 79.4 | 50.5 | 66.6 |
| STF | 1.9 | 61.8 | 70.8 | 56.7 | 86.3 | 64.9 | 57.3 | 77.8 | 49.5 | 65.6 |
| MBTF + STF | 1.9 | 61.9 | 70.5 | 57.4 | 86.0 | 66.2 | 58.9 | 78.1 | 51.1 | 66.3 |

## 4.3 COMPARISON TO TRAINING-FREE TOKEN REDUCTION METHODS

We further compare our method with training-free token reduction methods with 75% vision token reduction. Experimental results in Table 4 demonstrate that our method significantly outperforms training-free token reduction methods in all 6 popular vision-language benchmarks.

Table 4: Performance comparison to training-free token reduction methods.

| Method | GQA | SQA | VQA$^T$ | POPE | MMB | VQA$^{v2}$ |
|---|---|---|---|---|---|---|
| VisionZip | 57.6 | 68.9 | 56.8 | 80.5 | 62.0 | 75.6 |
| SparseVLM | 56.0 | 67.1 | 54.9 | 62.0 | 60.0 | 73.8 |
| MMTok | 59.3 | 69.0 | 57.0 | 85.8 | 62.3 | 76.4 |
| MustDrop | 56.9 | 68.5 | 56.3 | - | 61.1 | 74.6 |
| STF (ours) | **61.9** | **70.5** | **57.4** | **86.0** | **66.2** | **78.1** |

## 4.4 ABLATION STUDY

### 4.4.1 THE EFFECT OF FUSION MODULES

To substantiate the effectiveness of our fusion modules, we ablate each fusion module and compare their performance on 8 vision-language reasoning benchmarks.

As Table 3, LLaVA-1.5 with only MBTF significantly surpasses the baseline LLaVA-1.5 by 1.1% on the average score of 8 benchmarks. It implicates that features from previous layers of vision encoder contribute to the downstream vision-language tasks. Furthermore, we only evaluate the performance of LLaVA-1.5 with only STF, which achieves comparable performance as the original LLaVA-1.5, using only 25% vision tokens. The results demonstrate that excessive spatial redundancy exists in the sequence of vision tokens. Finally, we combine MBTF and STF to obtain more compact vision tokens. It achieves 0.7% performance gain based on the model with only STF, but doesn't outperform the model with only MBTF. The results are reasonable, since the model with MBTF and STF only costs 25% FLOPs of the one with MBTF. Overall, our proposed fusion modules can effectively improve the compactness of vision tokens and accelerate the inference without obvious performance drop.

### 4.4.2 THE EFFECT OF FUSION PARAMETERS

To explore optimal hyperparameters for token fusion, we compare the performance of our method under different kernel size $k$ and different numbers of fused tokens $E$. As results reported in Table 5, our method with $k = 2$ and $E = 1$ achieves the best performance. We find that more redundancy of vision tokens can not improve the performance of LLaVA model. Our method with $k = 2$ and $E = 2$ also outperforms the original LLaVA, where our method only uses 50% tokens of the original one. However, with the increment of kernel size, the performance also obviously drops. We speculate the potential reason as follows. The number of SFC parameters increases with kernel size, and they are randomly initialized. Due to the deficit of training data, the model overfits the dataset and doesn't achieve better performance.

### 4.4.3 THE EFFECT OF FUSION STRATEGIES

We also research different fusion strategies to reduce the length of vision tokens, which are fed into LLM. The first strategy is AvgPool, which directly averages adjacent $2 \times 2$ vision tokens as one

Table 5: The effect of fusion kernel size $k$ and #fused tokens $E$. All results are evaluated on the baseline of LLaVA-1.5 with Vicuna-1.5-7B. "Avg." denotes the average score of the previous 8 vision-language benchmarks.

| kernel size $k$ | #fused tokens $E$ | TFLOPs | GQA | SQA | VQA$^T$ | POPE | MMB | MMB$^{CN}$ | VQA$^{v2}$ | VisWiz | Avg. (%) |
|---|---|---|---|---|---|---|---|---|---|---|---|
| 1 | 1 | 7.6 | 62.0 | 66.8 | 58.2 | 85.9 | 64.3 | 58.3 | 78.5 | 50.0 | 65.5 |
| 2 | 1 | 1.9 | 61.9 | 70.5 | 57.4 | 86.0 | 66.2 | 58.9 | 78.1 | 51.1 | 66.3 |
| 2 | 2 | 3.8 | 62.7 | 69.1 | 56.2 | 86.2 | 65.6 | 58.8 | 77.9 | 51.1 | 66.0 |
| 4 | 4 | 1.9 | 61.2 | 69.4 | 52.0 | 84.8 | 64.2 | 57.2 | 77.1 | 42.3 | 63.5 |
| 4 | 8 | 3.8 | 60.4 | 70.1 | 53.2 | 84.6 | 64.9 | 57.5 | 77.2 | 46.8 | 64.3 |
| 8 | 4 | 0.5 | 35.3 | 37.2 | 29.4 | 59.6 | 43.2 | 36.8 | 52.1 | 24.9 | 39.8 |
| 8 | 8 | 0.9 | 52.1 | 57.6 | 41.2 | 75.8 | 54.2 | 46.2 | 60.2 | 33.5 | 52.6 |
| 8 | 16 | 1.9 | 58.6 | 69.2 | 49.8 | 83.3 | 63.4 | 54.4 | 75.8 | 42.5 | 62.1 |
| 8 | 32 | 3.8 | 59.2 | 68.6 | 48.9 | 83.5 | 62.7 | 54.1 | 75.6 | 39.2 | 61.5 |

Table 6: The effect of fusion strategies. All results are evaluated on the baseline of LLaVA-1.5 with Vicuna-1.5-7B. "Avg." denotes the average score of the previous 8 vision-language benchmarks.

| Method | TFLOPs | GQA | SQA | VQA$^T$ | POPE | MMB | MMB$^{CN}$ | VQA$^{v2}$ | VisWiz | Avg. (%) |
|---|---|---|---|---|---|---|---|---|---|---|
| AvgPool | 1.9 | 62.1 | 69.8 | 55.5 | 86.3 | 65.0 | 57.9 | 77.0 | 44.0 | 64.7 |
| TokenConcat | 1.9 | 62.5 | 69.7 | 56.1 | 85.9 | 66.2 | 59.2 | 77.8 | 46.3 | 65.4 |
| Q-Former | 1.9 | 60.4 | 68.9 | 56.8 | 84.3 | 64.2 | 57.4 | 77.1 | 49.6 | 64.8 |
| **STF (ours)** | 1.9 | 61.9 | 70.5 | 57.4 | 86.0 | 66.2 | 58.9 | 78.1 | 51.1 | 66.3 |

token. The second strategy is TokenConcat, which simply concatenates adjacent $2 \times 2$ vision tokens as one token. The third strategy is Q-Former (Li et al., 2023a), which conducts cross attention to fuse vision tokens into fewer learnable query tokens for token reduction. The results of these strategies are reported in Table 6. TokenConcat outperforms AvgPool by 0.7% average score on 8 vision-language benchmarks, yet inferior to our method. Compared to AvgPool, TokenConcat can reduce the information loss of token fusion, thus achieving better performance. Moreover, our method can adaptively fuse the vision tokens, thereby outperforming both AvgPool and TokenConcat.

### 4.4.4 THE EFFECT OF FUSION VISION BLOCKS

We compare the performance of different numbers of selected blocks for fusion in MBTF module, including 8, 16 and 24 blocks (evenly sampled). The results in Table 7 reveal that performance gain by fusing features from more blocks is not obvious. When visual features from 16 blocks are fused, its performance gain over 6 blocks is only 0.1% average accuracy. Fusing visual features from more blocks even leads to a significant performance degradation.

Table 7: The effect of fusion vision blocks.

| Method | GQA | SQA | VQA$^T$ | POPE | MMB | MMB$^{CN}$ | VQA$^{v2}$ | VisWiz | Avg. (%) |
|---|---|---|---|---|---|---|---|---|---|
| STF (ours, 8 blocks) | 61.9 | 70.5 | 57.4 | 86.0 | 66.2 | 58.9 | 78.1 | 51.1 | 66.3 |
| STF (ours, 16 blocks) | 62.1 | 70.4 | 57.8 | 86.2 | 66.4 | 59.0 | 77.9 | 51.0 | 66.4 |
| STF (ours, 24 blocks) | 61.7 | 70.1 | 57.1 | 85.8 | 66.0 | 58.3 | 77.3 | 50.2 | 65.8 |

### 4.5 PERFORMANCE ON OCR TASKS

For fine-grained vision-language tasks, we validate our method on OCR-related tasks, including OCRBench (Liu et al., 2024d) and ChartQA (Masry et al., 2022) benchmarks. Visionthink (Yang et al., 2025b) and VTC-Bench (Liao et al., 2025) claim that simple image downsampling for LMMs causes obvious performance degradation. However, the results shown in Table 8 reveal that our method can well address the above issue and achieve comparable performance as the original model on OCR tasks.

Table 8: Performance on OCR benchmarks.

| Method | OCRBench | ChartQA |
|---|---|---|
| LLaVA-1.5-7B (Liu et al., 2024a) | 297 | 18.2 |
| STF -7B (ours, $k = 2, E = 1$) | 301 | 18.1 |

### 4.6 INFERENCE LATENCY

In this section, we report the latency of our trained model for comprehensive evaluation of inference efficiency. Since the length of output texts vary for different models or high random, we follow the

settings of FastVLM (Vasu et al., 2025) and report average latency time to first token (per sample) using a single A100 GPU with batch size of 16 on VQAv2 benchmark. The results in Table 9 demonstrate that our method achieves $3.3\times$ speedup ratio on average latency time to first token over the original model.

Table 9: Latency evaluation of model inference.

| Model | LLaVA-1.5-7B | STF -7B (ours, $k = 2$, $E = 1$) | Speedup |
|---|---|---|---|
| **Average Latency (ms)** | 346.2 | 103.8 | $3.3\times$ |

## 4.7 EVALUATION ON LARGER LLM

For more model sizes, we validate our method on LLaVA-1.5-13B model with full parameter finetuning. We follow the training hyper-parameters of LLaVA-1.5-13B and token fusion hyper-parameters of STF ($k = 2$, $E = 1$, retain 25% tokens). The experimental results in Table 10 show that our method achieves comparable performance on large LLaVA-1.5 model with only 25% vision tokens.

Table 10: Performance on Large LLM.

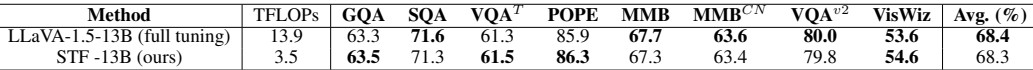

| Method | TFLOPs | GQA | SQA | VQA$^T$ | POPE | MMB | MMB$^{CN}$ | VQA$^{v2}$ | VisWiz | Avg. (%) |
|---|---|---|---|---|---|---|---|---|---|---|
| LLaVA-1.5-13B (full tuning) | 13.9 | 63.3 | **71.6** | 61.3 | 85.9 | **67.7** | **63.6** | **80.0** | 53.6 | **68.4** |
| STF -13B (ours) | 3.5 | **63.5** | 71.3 | **61.5** | **86.3** | 67.3 | 63.4 | 79.8 | **54.6** | 68.3 |

## 4.8 EVALUATION ON DIFFERENT IMAGE RESOLUTIONS

To validate the effectiveness of our method on different image resolutions, we train our model by images with resolution of $224 \times 224$ and $336 \times 336$. The settings follow LLaVA-1.5-7B. The results in Table 11 support the effectiveness of our method on different input image resolutions.

Table 11: Evaluation on Different Image Resolutions

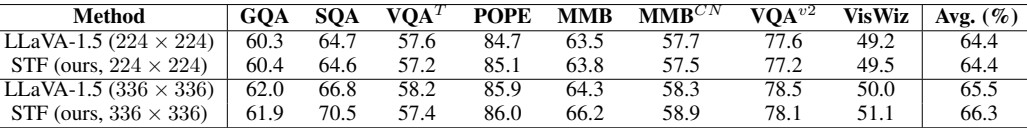

| Method | GQA | SQA | VQA$^T$ | POPE | MMB | MMB$^{CN}$ | VQA$^{v2}$ | VisWiz | Avg. (%) |
|---|---|---|---|---|---|---|---|---|---|
| LLaVA-1.5 ($224 \times 224$) | 60.3 | 64.7 | 57.6 | 84.7 | 63.5 | 57.7 | 77.6 | 49.2 | 64.4 |
| STF (ours, $224 \times 224$) | 60.4 | 64.6 | 57.2 | 85.1 | 63.8 | 57.5 | 77.2 | 49.5 | 64.4 |
| LLaVA-1.5 ($336 \times 336$) | 62.0 | 66.8 | 58.2 | 85.9 | 64.3 | 58.3 | 78.5 | 50.0 | 65.5 |
| STF (ours, $336 \times 336$) | 61.9 | 70.5 | 57.4 | 86.0 | 66.2 | 58.9 | 78.1 | 51.1 | 66.3 |

## 4.9 EVALUATION ON LLAMA-3.1 LLM

We further validate our method on Llama-3.1-8B backbone. We adopt the default settings as LLaVA-1.5-7B to pretrain and finetune the model. The results in Table 12 demonstrate that our method with Llama-3.1-8B backbone achieves comparable performance with only 25% vision tokens, which supports the effectiveness of our method on other LLM backbones.

Table 12: Evaluation on Other LLM Architecture

| Method | GQA | SQA | VQA$^T$ | POPE | MMB | MMB$^{CN}$ | VQA$^{v2}$ | VisWiz | Avg. (%) |
|---|---|---|---|---|---|---|---|---|---|
| LLaVA (Llama-3.1-8B) | 62.4 | 70.9 | 58.1 | 86.1 | 65.9 | 59.9 | 78.8 | 52.1 | 66.8 |
| STF (ours, Llama-3.1-8B) | 62.2 | 71.1 | 57.8 | 86.5 | 66.8 | 59.4 | 78.9 | 51.7 | 66.8 |

## 5 CONCLUSION AND FUTURE WORK

In this paper, we propose a novel token fusion method to reduce vision tokens fed into LLM, thereby accelerating the inference of LMM. To this end, we introduce Multi-Block Token Fusion and Spatial Token Fusion module to fuse multi-granularity representations of vision encoder and reduce spatial redundancy. The experimental results demonstrate that our method with only 25% vision tokens achieves comparable or even superior performance to the baseline. It indeed supports that our method can effectively reduce the redundancy of vision token sequence, while maintaining the information of original images. In future work, we plan to fuse vision tokens according to the context of text tokens for more efficient vision token reduction.

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

# A APPENDIX

## A.1 EVALUATION ON LEARNABLE VISION ENCODER

Due to the small dataset size for the training, it's challenging for us to training all parameter as joint finetuning. To this end, we insert the vision encoder finetuning stage between projection pretraining and projection-LLM joint finetuning stage to further boost the performance. The experimental results in Table 13 demonstrate that our method can further improve performance on vision-language benchmarks, while achieving consistent efficiency as the one with frozen vision encoder.

Table 13: Comparison to adaptive tokenization method

| Method | GQA | SQA | $VQA^T$ | POPE | MMB | $MMB^{CN}$ | $VQA^{v2}$ | VisWiz | Avg. (%) |
|---|---|---|---|---|---|---|---|---|---|
| STF (frozen vision encoder) | 61.9 | 70.5 | **57.4** | 86.0 | 66.2 | **58.9** | 78.1 | 51.1 | 66.3 |
| STF (tuning vision encoder) | **62.2** | **70.8** | 57.2 | **86.6** | **66.7** | 58.6 | **78.4** | **51.5** | **66.5** |

## A.2 THE DISCUSSION ABOUT ADDITIONAL COST OF MBTF

Since the parameter number of our projection modules, including MBTF (37.7 million parameters) and STF (134.2 million parameters), is significantly smaller than the one of LLM (7 billion parameters), the additional cost introduced by the fine-tuning stage only takes a very small portion of total fine-tuning cost.

We further compare average fine-tuning time of mini-batch (batch_size = 128, 100 mini-batch iterations, 8 RTX A6000 GPUs) between our method and the original LLaVA-1.5-7B model. The average fine-tuning time of our method is 18.6s per iteration, while the one of LLaVA-1.5-7B model is 27.8s per iteration. Because our method introduces fewer vision tokens after fusion for the training of LLM, our method even achieves significantly higher fine-tuning efficiency.

## A.3 COMPARISON TO ADAPTIVE TOKENIZATION METHOD

For adaptive tokenization method, we apply ElasticTok-VAE (Yan et al., 2024) on LLaVA-1.5-7B model to reduce the number of vision token and compare with our method. The results in Table 14 show that our method significantly outperforms the counterpart with ElasticTok-VAE.

Table 14: Comparison to adaptive tokenization method

| Method | GQA | SQA | VQA$^T$ | POPE | MMB | MMB$^{CN}$ | VQA$^{v2}$ | VisWiz | Avg. (%) |
|---|---|---|---|---|---|---|---|---|---|
| ElasticTok-VAE | 60.6 | 69.2 | 56.9 | 84.7 | 64.6 | 57.6 | 77.3 | 49.9 | 65.1 |
| STF (ours) | 61.9 | 70.5 | 57.4 | 86.0 | 66.2 | 58.9 | 78.1 | 51.1 | 66.3 |

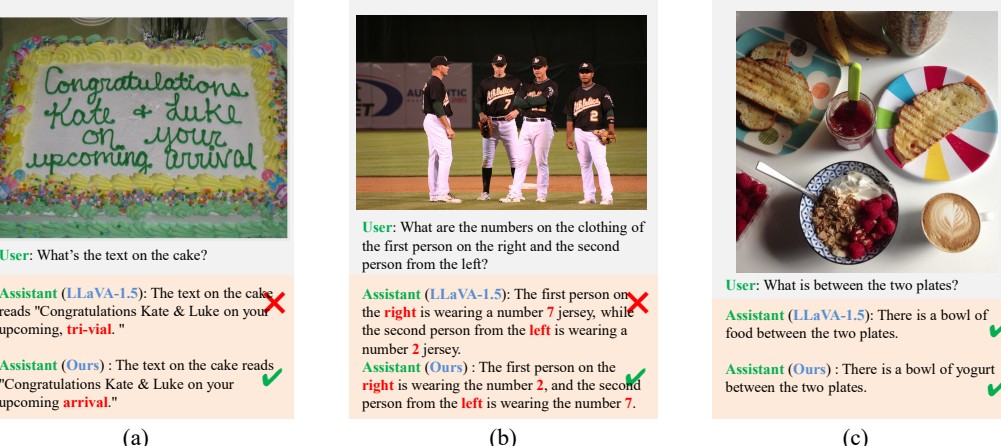

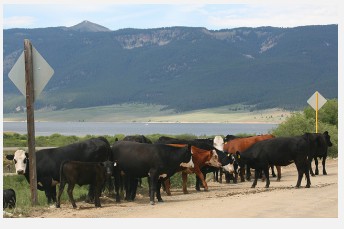

Figure 3: Case study of LLaVA-1.5 and our proposed method.

## A.4 CASES STUDY

To better understand the property of our method, we present some specific cases. As shown in Figure 3, we compare our method with the baseline LLaVA-1.5 with Vicuna-1.5-7B. In Figure 3 (a), our method successfully recognize the last word "arrival", but LLaVA-1.5 mistakes it as "tri-vial". This case shows that our method can well recognize the blurred image information by understanding the context of images. In Figure 3 (d), LLaVA-1.5 fails to recognize the number of bars, but understands the expenditure for fruits is lowest. On the contrary, our method smoothly identify both the number of bars and the lowest expenditure. Our method also achieves superior performance on the counting of cows in Figure 3 (d). The above cases reveal that our method can better understand image details, in spite of fewer vision tokens used. Overall, the above experimental results demonstrates that our method can effectively reduce the spatial redundancy, while maintaining the vision-language reasoning capabilities.

