# OpenReview forum: "Learning Compact Vision Tokens for Large Multimodal Models"
_ICLR.cc/2026/Conference — ICLR 2026 Conference Withdrawn Submission_

### Official Review · Reviewer_Gjz5 · 2025-10-27

**Soundness:** 2
**Presentation:** 2
**Contribution:** 2
**Rating:** 4
**Confidence:** 3

**Summary:**

This paper proposes a method for accelerating large multimodal models (LMMs) by substantially reducing the number of vision tokens supplied to the language model, while striving to maintain or enhance multimodal performance. The core contribution is a dual-stage token compression pipeline: (1) a Multi-Block Token Fusion (MBTF) module aggregates multi-level features from selected vision encoder layers, and (2) a Spatial Token Fusion (STF) module fuses spatially adjacent vision tokens into compact representations via learnable convolutions. Experiments on the LLaVA-1.5-7B backbone across eight popular vision-language benchmarks validate the approach, showing that with only 25% of the original vision tokens, the proposed method achieves comparable or improved performance versus the full-token baseline and several state-of-the-art efficient LMMs.

**Strengths:**

1. Clear Identification of Redundancy: The motivation—identifying excessive spatial redundancy in standard vision token sequences for LMMs—is empirically substantiated in Figure 1 and corresponding baseline experiments, which demonstrate minor performance losses or even gains when aggressively pooling tokens.

2. Principled Dual-Stage Compression: The architecture integrates MBTF for multi-granularity feature fusion, followed by STF for spatially adaptive reduction, as clearly illustrated in Figure 2 and specified in detail with Table 1. This strategy is logical and well-justified by the observed redundancy in vision tokens.

**Weaknesses:**

## **1. Missing Coverage of Directly Related Recent Work**
A substantial body of recent, directly relevant work on token compression, adaptive visual tokenization, and information steering in multimodal models is omitted from both the related work (Section 2) and experimental comparisons:
Notable works missing include [Cheng et al., 2024] (survey), [Zhang et al., 2024] (MMTok, SEED, Multi-Stage Vision Token Dropping, ElasticTok), [Zhang et al., 2024] (VoCo-LLaMA), [Li et al., 2025] (Hidden Life of Tokens), etc.
The discussion focuses primarily on token pruning and merging from existing efficient LLaVA variants, but does not adequately position this method with respect to the broader, very recent multimodal token compression literature (see Potentially Missing Related Work section below for specific papers and integration suggestions). This oversight undermines both the novelty claims and the work's perceived significance and must be addressed for publication.

## **2. Limited Theoretical Justification for "Lossless" Claim**
The paper repeatedly claims "lossless" token reduction when $k=2$, based on token/channel dimension matching (Section 3.4). However, the rationale lacks rigorous theoretical support or formal information-theoretic analysis. The claim is based on the matching of hidden sizes, rather than provable metrics regarding preserved semantic content. For instance, Eqn (in Section 3.4) specifies the process, but does not analyze information bottleneck, nor the risk of over-/underfitting due to arbitrarily increasing channels. This may mislead readers regarding the true trade-off between compactness and fidelity.

## **3. Empirical Evidence not Comprehensive Enough**
While Table 2 presents an extensive comparison on eight benchmarks, direct experimental contrasts with many distinct token reduction strategies (e.g., methods selected from the missing related work above) are absent. The ablations (Tables 3–5) primarily compare in-house baselines (MBTF, STF, simple pooling), while more rigorous comparison to recent learning-free or adaptive tokenization methods would strengthen the empirical evidence. In the absence of such, it remains unclear if the proposed approach outperforms the current state-of-the-art across scenarios.

## **4. Component and Parameter Clarity Limitations**
The mathematical specification and parameterization of the fusion kernels and projection modules are described, but details on their initialization, regularization, and design choices (e.g., GeLU activations, precise effect of kernel stride/sizes in practice) are somewhat sparse. For instance, in Table 1 and Section 3.4, a rationale for channel size progression, and potential overfitting in larger-k models (mentioned in Section 4.3.2) is offered, but more systematic analysis or regularization ablation would improve transparency.

## **5. Unclear Generalization Across Modalities/Backbones**
The method is tested only on LLaVA-1.5-7B (with CLIP-ViT and Vicuna-1.5-7B). There is no demonstration of cross-backbone generality or performance on models using different visual architectures or non-Vicuna LLMs—limiting the generalizability of the claims. Additionally, experiments mostly cover high-level vision-language reasoning and VQA; more fine-grained domains (e.g., visual grounding or image captioning) are not evaluated, which would clarify the trade-off between compression and task specificity.

## **6. Ablation Study Depth**
The ablations, while illuminating, would be stronger if (i) out-of-distribution robustness (e.g., to images substantially different from training data) were reported, (ii) FLOP/token reductions were systematically related to accuracy for a broader set of $k/E$ values, and (iii) an explicit analysis of failure cases under aggressive token compression was included.

## **7. Ambiguity in Performance Claims for STF vs. MBTF**
Table 3 suggests MBTF alone slightly outperforms the combined STF+MBTF approach in outright accuracy, yet this is brushed aside as a trade-off for efficiency. More nuanced discussion of why the combination does not enhance average accuracy, and scenarios where one should be prioritized over the other, would enhance scientific rigor.

## **8. Minor Typographical and Expositional Issues**
There are instances of unclear phrasing and notation, e.g., "stride of 2, and only $25 %$ tokens are remained" (Page 1), or ambiguous variable definitions in some formulae—these do not prohibit comprehension, but do reduce polish.

## **9. Assumption of Vision Encoder Freezing**
The method relies on a frozen vision encoder throughout training (Section 3.5, 4.1.3). It is unclear whether similar efficiency/performance gains would be possible if joint tuning with visual features occurred. This assumption may limit applicability in scenarios where full model finetuning is desired.

## **10. Code release**
The authors claim that they will release the code in the future. However, I think the code is available to reviewers during this stage, and authors are suggested to upload the source code as the supplementary material.

**Questions:**

Please see the weaknesses

---

> ### Author Response · Authors · 2025-11-17
> **Request for the missing literature information at Weakness 1**
>
> Dear Reviewer Gjz5,
>
> Regarding Weakness 1, you mentioned that we missed some relevant literature.
>
> Could you please provide the specific titles or authors of these works?
>
> We were unable to locate the exact papers based on the current description and would appreciate the details so we can cite and discuss them properly in our revision.
>
> Authors

---

> > ### Comment · Reviewer_Gjz5 · 2025-11-17
> > **References for the weakness 1**
> >
> > Hi, I apologize for missing the references for weakness 1, and you can find the references below.
> >
> > [1] Shao K, Tao K, Zhang K, et al. When tokens talk too much: A survey of multimodal long-context token compression across images, videos, and audios[J]. arXiv preprint arXiv:2507.2019.
> >
> > [2] Ye X, Gan Y, Huang X, et al. Voco-llama: Towards vision compression with large language models[C]//Proceedings of the Computer Vision and Pattern Recognition Conference. 2025: 29836-29846.
> >
> > [3] Li Z, Shi H, Gao Y, et al. The hidden life of tokens: Reducing hallucination of large vision-language models via visual information steering[J]. arXiv preprint arXiv:2502.03628, 2025.
> >
> > [4] Dong S, Hu J, Zhang M, et al. MMTok: Multimodal Coverage Maximization for Efficient Inference of VLMs[J]. arXiv preprint arXiv:2508.18264, 2025.
> >
> > [5] Yu G, Chen Y, Xu J. Sparsity Meets Similarity: Leveraging Long-Tail Distribution for Dynamic Optimized Token Representation in Multimodal Large Language Models[J]. arXiv preprint arXiv:2409.01162, 2024.
> >
> > [6] Zhao Z, Li Y, Li Y. Learning Free Token Reduction for Multi-Modal LLM[J]. arXiv e-prints, 2025: arXiv: 2501.17391.
> >
> > [7] Ge Y, Ge Y, Zeng Z, et al. Planting a SEED of vision in large language model. arXiv 2023[J]. arXiv preprint arXiv:2307.08041.
> >
> > [8] Liu T, Shi L, Hong R, et al. Multi-stage vision token dropping: Towards efficient multimodal large language model[J]. arXiv preprint arXiv:2411.10803, 2024.
> >
> > [9] Yan W, Mnih V, Faust A, et al. Elastictok: Adaptive tokenization for image and video[J]. arXiv preprint arXiv:2410.08368, 2024.
> >
> > Hope these references can help you solve my concerns, and I will raise my score to 6 if you can address my concerns well.
> >
> > Best
> >
> > Reviewer Gjz5

---

> ### Author Response · Authors · 2025-11-20
>
> Thanks for your helpful feedback. We address your concerns as follows.
>
> - **The supplement of related work about token compression and adaptive visual tokenization.**
>
>   Thank you for pointing out the missing references.
>
>   We supplement the discussion about the missing references in **related work section of revised paper**. Comparative experiments can be found in the following discussion.
>
> - **Theoretical justification about lossless claim.**
>
>   Actually, establishing the effectiveness of deep learning methodologies through rigorous theoretical justification **remains a formidable and unresolved challenge within the academic community**. For example, it is challenging to rigorously justification from a theoretical standpoint whether ViTs or CNNs are definitively superior. **Currently, conclusions in the field of deep learning rely predominantly on empirical evidence rather than on rigorous theoretical justification **.
>
>   Similarly, **the design of our method is based on intuitive analysis and empirical study**. Our intuitive idea is that fused vision token fed into LLM, which is identical to the one of adjacent $2\times2$ vision tokens produced by vision encoder, is benefit to achieve lossless token reduction. The empirical results indeed support the effectiveness of our claim.
>
> - **Comparison to recent learning-free or adaptive tokenization methods.**
>
>   We compare our method with with recent training-free compression approaches as follows.
>
>   |     Method     |   GQA    |   SQA    |   VQAT   |   POPE   |   MMB    |  VQAv2   |
>   | :------------: | :------: | :------: | :------: | :------: | :------: | :------: |
>   |   MMTok [4]    |   59.3   |   69.0   |   57.0   |   85.8   |   62.3   |   76.4   |
>   |  MustDrop [8]  |   56.9   |   68.5   |   56.3   |    -     |   61.1   |   74.6   |
>   | **STF (ours)** | **61.9** | **70.5** | **57.4** | **86.0** | **66.2** | **78.1** |
>
>   Experimental results demonstrate that our method achieves significantly better performance than training-free compression approaches.
>
>   We supplement the above results in **Section 4.3 of revised paper**.
>
>   For adaptive tokenization method, we apply ElasticTok-VAE [9] on LLaVA-1.5-7B model to reduce the number of vision token and compare with our method as below.
>
>   |        Method         | GQA  | SQA  | VQAT | ROPE | MMB  | MMBCN | VQAv2 | VisWiz | Avg  |
>   | :-------------------: | :--: | :--: | :--: | :--: | :--: | :---: | :---: | :----: | :--: |
>   | LLaVA-1.5-7B with [9] | 60.6 | 69.2 | 56.9 | 84.7 | 64.6 | 57.6  | 77.3  |  49.9  | 65.1 |
>   |    **STF (ours)**     | 61.9 | 70.5 | 57.4 | 86.0 | 66.2 | 58.9  | 78.1  |  51.1  | 66.3 |
>
>   The results show that our method significantly outperforms the counterpart with ElasticTok-VAE. We supplement the above results in **Section A.3 of revised paper appendix**.
>
> - **More details about design choices.**
>
>   Thanks for your helpful comments.
>
>   We supplement hyper-parameters of design choices in Section 4.4.2 as below. All experiments in Table 4 of Section 4.4.2 use stride size identical to kernel size. The number of fused tokens $E$ corresponds to the output channel size of the first convolution layer of STF module. For example, when $k=8$ and $E=32$, the output channel size of the first convolution layer of STF module is (batch_size, height, width, 32$\times$4096), then reshaped into (batch_size, 32, height, width, 4096) as 32 fused tokens as output. In other words, our STF module fuses $8\times8=64$ tokens into 32 fused tokens and then feeds them into LLM.

---

> ### Author Response · Authors · 2025-11-20
>
> - **Experiments on other LLMs and more fine-grained domain.**
>
>   We further validate our method on Llama-3.1-8B as LLM backbone. We adopt the default settings as LLaVA-1.5-7B to pretrain and finetune the model. The results are reported as follows.
>
>   |            Method             | GQA  | SQA  | VQAT | ROPE | MMB  | MMBCN | VQAv2 | VisWiz | Avg  |
>   | :---------------------------: | :--: | :--: | :--: | :--: | :--: | :---: | :---: | :----: | :--: |
>   |     LLaVA (Llama-3.1-8B)      | 62.4 | 70.9 | 58.1 | 86.1 | 65.9 | 59.9  | 78.8  |  52.1  | 66.8 |
>   | **STF** ( ours, Llama-3.1-8B) | 62.2 | 71.1 | 57.8 | 86.5 | 66.8 | 59.4  | 78.9  |  51.7  | 66.8 |
>
>   The above results demonstrate that our method with Llama-3.1-8B backbone achieves comparable performance   with only 25% vision tokens, which supports the effectiveness of our method on other LLM backbones.
>
>   We supplement the above results in **Section 4.9 of revised paper**.
>
>   For **evaluation on fine-grained tasks**, we choose OCR benchmarks, including OCRBench and ChartQA, and report the results as below.
>
>   | Method                       | OCRBench | ChartQA |
>   | :--------------------------- | :------: | :-----: |
>   | LLaVA-1.5-7B                 |   297    |  18.2   |
>   | STF -7B (ours, $k=2$, $E=1$) |   301    |  18.1   |
>
>   The results demonstrate that our method achieves comparable performance on resolution-sensitive benchmarks, OCRBench and ChartQA.
>
>   We supplement the above results in **Section 4.5 of revised paper**.
>
> - **Experiments on OOD dataset, FLOP/token reduction on broader set of k/E values, analysis of failure cases.**
>
>   For out-of-distribution robustness, we evaluate our method on OCR benchmarks, including OCRBench and ChartQA. The results in the above table supports the robustness of our method on out-of-distribution data.
>
>   We supplement more results with various E values when k=8 as follows.
>
>   | k    |  E   | TFLOPs | #Tokens | GQA  | SQA  | VQAT | ROPE | MMB  | MMBCN | VQAv2 | VisWiz | Avg  |
>   | ---- | :--: | :----: | :-----: | :--: | :--: | :--: | :--: | :--: | :---: | :---: | :----: | :--: |
>   | 8    |  4   |  0.5   |   36    | 35.3 | 37.2 | 29.4 | 59.6 | 43.2 | 36.8  | 52.1  |  24.9  | 39.8 |
>   | 8    |  8   |  0.9   |   72    | 52.1 | 57.6 | 41.2 | 75.8 | 54.2 | 46.2  | 60.2  |  33.5  | 52.6 |
>   | 8    |  16  |  1.9   |   144   | 58.6 | 69.2 | 49.8 | 83.3 | 63.4 | 54.4  | 75.8  |  42.5  | 62.1 |
>   | 8    |  32  |  3.8   |   288   | 59.2 | 68.6 | 48.9 | 83.5 | 62.7 | 54.1  | 75.6  |  39.2  | 61.5 |
>
>   Compared to k=2 and k=4, the performance of k=8 decreases dramatically. We speculate that the large fusion kernel significantly increases the hardness of token fusion. In future work, we plan to explore token fusion strategy with larger kernel for high compression ratio.
>
>   We supplement the above results in **Section 4.4.2 of revised paper**.
>
> - **Discussion about the combination of STF and MBTF.**
>
>   We analyze the reason why the combination of STF and MBTF doesn't enhance average accuracy as follows. It's because that STF reduces the number of vision tokens from 100% to only 25%. But MBTF uses all vision tokens produced by vision encoder, thus achieving better average accuracy. It's designed for the trade-off between performance and efficiency. When extreme performance is required, we can only apply MBTF on LMMs. When we need the balance between performance and efficiency, we choose the combination of STF and MBTF.
>
> - **Typographical and expositional issues.**
>
>   We have fixed them in the revised paper.
>
> - **The effeciency/performance gain if joint tuning with visual features occurred.**
>
>   Due to the small dataset size for the training, it's challenging for us to training all parameter as joint finetuning. To this end, we insert the vision encoder finetuning stage between projection pretraining and projection-LLM joint finetuning stage to further boost the performance. The results are reported as follows.
>
>   |             Method              | GQA  | SQA  | VQAT | ROPE | MMB  | MMBCN | VQAv2 | VisWiz | Avg  |
>   | :-----------------------------: | :--: | :--: | :--: | :--: | :--: | :---: | :---: | :----: | :--: |
>   | **STF** (frozen vision encoder) | 61.9 | 70.5 | 57.4 | 86.0 | 66.2 | 58.9  | 78.1  |  51.1  | 66.3 |
>   | **STF** (tuning vision encoder) | 62.2 | 70.8 | 57.2 | 86.6 | 66.7 | 58.6  | 78.4  |  51.5  | 66.5 |
>
>   The experimental results demonstrate that our method can further improve performance on the above 8 benchmarks, while achieving consistent efficiency as the one with frozen vision encoder.
>
>   We supplement the above results in **Section A.1 of revised paper appendix**.
>
> - **Code release.**
>
>   We have uploaded our code as supplementary material.

---

> > ### Comment · Reviewer_Gjz5 · 2025-11-21
> > **Response to the rebuttal**
> >
> > Thanks for the authors' reply. Overall, I am satisfied with this response and want to further discuss with the authors.
> >
> > 1) STF performs the token reduction only on the visual features. However, such a token reduction method may lose some visual features related to the text token. Whether the future work can combine the text tokens to reduce the visual tokens? Maybe this way can help select the key visual tokens that the text token really focuses on.
> >
> > 2) If the authors have enough time, I suggest that the comparison with the following works can be added to Table 6: TokenPacker [1] and MiniGemini [2]. These two methods also perform visual token reduction based on the features from different blocks.
> >
> > [1] Li W, Yuan Y, Liu J, et al. Tokenpacker: Efficient visual projector for multimodal llm[J]. International Journal of Computer Vision, 2025: 1-19.
> >
> > [2] Li Y, Zhang Y, Wang C, et al. Mini-gemini: Mining the potential of multi-modality vision language models[J]. arXiv preprint arXiv:2403.18814, 2024.
> >
> > **I will raise my score to 6**. Good luck!
> >
> > Best
> >
> > Reviewer Gjz5

---

> ### Author Response · Authors · 2025-11-22
>
> Thanks for your timely and helpful response.
>
> We address your remaining concerns as below.
>
> - This comment provides us a very promising research line in future work and we supplement the discussion about it in our revised conclusion section.
>
> - We add the discussion about TokenPacker and MiniGemini in related work section of our revised paper. However, due to the multi-scale images (including higher resolution images, more than $336 \times 336$) used in their method and experiments, our method trained by only single $336 \times 336$ resolution images can not be directly compare with them fairly.
>
> Could you raise the rating score to support our work?
> Thank you very much.

---

> > ### Comment · Reviewer_Gjz5 · 2025-11-24
> > **Response to the further discussion**
> >
> > I think the author's latest response was somewhat perfunctory as follows:
> >
> > 1) For point 1, my intention was to discuss with the author whether it's possible to combine text features to reduce visual tokens, and I do not require the author to supplement the above discussion in the paper. However, authors just respond to me with only "further work". This is rather disappointing.
> >
> > 2) For point 2, the reason for comparison with Tokenpacker and Mini-Gemini is that I am curious about the influence of the multi-scale feature. If authors do not have enough time, you can point out your opinions about whether the multi-scale feature is effective for your method. Besides, where can I find the discussion about TokenPacker and MiniGemini in your paper?
> >
> > I believe the rebuttal stage also includes discussion, rather than simply offering perfunctory responses to reviewers to gain a higher score. In the current stage, **I think the score of 6 (borderline) is suitable for this paper**.
> >
> > Best
> >
> > Reviewer Gjz5

---

> > > ### Author Response · Authors · 2025-12-03
> > >
> > > Thanks for your response.
> > > - We indeed response your feedback about `Whether the future work can combine the text tokens to reduce the visual tokens?`.
> > > - We indeed discuss TokenPacker and MiniGemini in the **related work** section of our revised paper.

---

### Official Review · Reviewer_kah7 · 2025-10-30

**Soundness:** 3
**Presentation:** 2
**Contribution:** 2
**Rating:** 6
**Confidence:** 3

**Summary:**

The paper proposes a two-stage projector for large multimodal models (LMMs) that reduces the number of vision tokens fed to the large language model (LLM) while preserving task performance. First, a Multi-Block Token Fusion (MBTF) module concatenates intermediate features from evenly sampled visual blocks and compresses them with convolutions. Second, a Spatial Token Fusion (STF) module applies another convolution that fuses adjacent spatial tokens, followed by alignment to the LLM embedding space through convolutions. Experiments show that the proposed approach matches or exceeds some prior baselines on the LLaVA-1.5 model under 1.9 TFLOPs.

**Strengths:**

* The paper is clearly written and easy to follow.

* Thorough ablation studies are provided, including fusion modules, kernel sizes, and fusion strategies, as well as qualitative case studies, which help readers gain a better understanding of how the proposed approach functions and contributes to overall performance.

**Weaknesses:**

* End-to-end latency metrics are missing. Since lower TFLOPs don’t necessarily lead to significantly faster wall-clock speed, I’d suggest that the authors report latency to better assess efficiency improvements.

* It seems all experiments are conducted on LLaVA-1.5-7B with only one size and one input resolution. It is important to verify the effectiveness of the proposed method in more diverse settings.

* The additional cost introduced by the fine-tuning stage is not clearly discussed. The paper indicates that both the LLM and the projector are fine-tuned, which may involve notable computational expense.

* In terms of token reduction on LLaVA, some related work, such as [1], is not discussed in the paper. Clarifying how the proposed method relates to or differs from this prior line of work would be helpful for readers. [1] CrossGET: Cross-Guided Ensemble of Tokens for Accelerating Vision-Language Transformers by Shi et al. In ICML24.

**Questions:**

Please refer to the Weaknesses Section.

---

> ### Author Response · Authors · 2025-11-20
>
> We appreciate your positive feedback. We handle your concerns as follows.
>
> - **The results on end-to-end latency metrics.**
>
>   Thanks for very constructive suggestion.
>
>   Since the length of output texts vary for different models or high random, we follow the settings of FastVLM [1] and report average latency time to first token (per sample) using a single A100 GPU with batch size of 16 on VQAv2 benchmark. The results are listed as below.
>
>   |        Model         | LLaVA-1.5-7B | STF -7B (ours, $k=2$, $E=1$) |   Speedup   |
>   | :------------------: | :----------: | :--------------------------: | :---------: |
>   | Average Latency (ms) |    346.2     |            103.8             | 3.3$\times$ |
>
>   The above results demonstrate that our method achieves 3.3$\times$ speedup ratio on average latency time to first token over the original model.
>
>   We supplement the above results in **Section 4.6 of revised paper**.
>
>   [1] FastVLM: Efficient Vision Encoding for Vision Language Models. CVPR 2025.
>
> - **Experiments on more model sizes and input resolutions.**
>
>   Thanks for your suggestion.
>
>   For more model sizes, we validate our method on LLaVA-1.5-13B model with full parameter finetuning. We follow the training hyper-parameters of LLaVA-1.5-13B and token fusion hyper-parameters of STF ($k=2$, $E=1$, retain 25% tokens). The results are reported as follows.
>
>   |           Method            |   GQA    |   SQA    |   VQAT   |   ROPE   |   MMB    |  MMBCN   |  VQAv2   |  VisWiz  |   Avg    |
>   | :-------------------------: | :------: | :------: | :------: | :------: | :------: | :------: | :------: | :------: | :------: |
>   | LLaVA-1.5-13B (full tuning) |   63.3   | **71.6** |   61.3   |   85.9   | **67.7** | **63.6** | **80.0** | **53.6** | **68.4** |
>   |     **STF -13B (ours)**     | **63.5** |   71.3   | **61.5** | **86.3** |   67.3   |   63.4   |   79.8   | **54.6** |   68.3   |
>
>   The above results demonstrate that our method on LLaVA-1.5-13B achieves comparable performance as the original LLaVA-1.5-13B with only 25% vision tokens. We supplement the above results in **Section 4.7 of revised paper**.
>
>   For more input resolutions, we validate our method on both input resolution of $224 \times 224$ and $336 \times 336$. The settings follow LLaVA-1.5-7B. The results are listed as below.
>
>   |              Method               | GQA  | SQA  | VQAT | ROPE | MMB  | MMBCN | VQAv2 | VisWiz | Avg  |
>   | :-------------------------------: | :--: | :--: | :--: | :--: | :--: | :---: | :---: | :----: | :--: |
>   |   LLaVA-1.5 ($224 \times 224$)    | 60.3 | 64.7 | 57.6 | 84.7 | 63.5 | 57.7  | 77.6  |  49.2  | 64.4 |
>   | **STF** (ours, $224 \times 224$)  | 60.4 | 64.6 | 57.2 | 85.1 | 63.8 | 57.5  | 77.2  |  49.5  | 64.4 |
>   |   LLaVA-1.5 ($336 \times 336$)    | 62.0 | 66.8 | 58.2 | 85.9 | 64.3 | 58.3  | 78.5  |  50.0  | 65.5 |
>   | **STF** ( ours, $336 \times 336$) | 61.9 | 70.5 | 57.4 | 86.0 | 66.2 | 58.9  | 78.1  |  51.1  | 66.3 |
>
>   The results shows that our method can also achieve effective vision token reduction when the input resolution is set to $224 \times 224$. We supplement the above results in **Section 4.8 of revised paper**.
>
> - **The discussion about the additional cost introduced by the fine-tuning stage.**
>
>   Since the parameter number of our projection modules, including MBTF (37.7 million parameters) and STF (134.2 million parameters),  is significantly smaller than the one of LLM (7 billion parameters), the additional cost introduced by the fine-tuning stage **only takes a very small portion of total fine-tuning cost**.
>
>   We further compare average fine-tuning time of mini-batch (batch_size = 128, 100 mini-batch iterations, 8 RTX A6000 GPUs) between our method and the original LLaVA-1.5-7B model. The average fine-tuning time of our method is **18.6s per iteration**, while the one of LLaVA-1.5-7B model is **27.8s per iteration**. Because our method introduces fewer vision tokens after fusion for the training of LLM, our method even achieves significantly higher fine-tuning efficiency.
>
>   We add the above discussion in **Section A.2 of revised paper appendix**.
>
> - **The supplement of related work about token reduction.**
>
>   Thank you for pointing out the missing important related work.
>
>   CrossGET [1] conducts cross-guided matching and ensemble to adaptively combine both vision and text tokens for the acceleration of LMMs. This method effectively improves inference efficiency of LMMs, while maintaining high performance.
>
>   We supplement the above discussion about CrossGET in **related work section of our revised paper**.
>
>   [1] Shi et al. CrossGET: Cross-Guided Ensemble of Tokens for Accelerating Vision-Language Transformers. ICML 2024.

---

### Official Review · Reviewer_k6si · 2025-11-04

**Soundness:** 2
**Presentation:** 2
**Contribution:** 2
**Rating:** 4
**Confidence:** 5

**Summary:**

The paper try to tackle LMM inference cost by compressing vision tokens. It introduces Spatial Token Fusion (STF) to merge spatially adjacent tokens and Multi-Block Token Fusion (MBTF) to re-inject multi-granularity features after reduction. Built on LLaVA-1.5, the method preserves accuracy on 8 VL benchmarks while using ~25% of baseline vision tokens, yielding speedups. Code and weights are promised for release.

**Strengths:**

1. The research question is interesting, and the scenario has clear practical relevance.

2. Across multiple benchmarks, the LLaVA-1.5–based approach delivers competitive performance.

**Weaknesses:**

1. **On Generalization vs. OCR Tasks**

   Papers [1] and [2] have shown that in many scenarios, even simple resizing techniques yield strong performance—consistent with this paper’s claim that AvgPool works well. However, both also highlight that on OCR-related tasks, aggressive token reduction often leads to notable performance degradation. I therefore recommend adding evaluations on OCR-specific benchmarks such as *OCRBench* and *ChartQA* to better assess robustness in those contexts.

2. **Applicability to Other VLMs**

   Can the proposed method be directly applied to models like *Qwen2.5-VL* [3] or *LLaVA-NeXT* [4]? Clarifying compatibility or adaptation requirements would strengthen the paper’s practical relevance.

3. **Outdated Baselines**

   The choice of baselines appears limited. Since the method is training-based, I suggest comparisons with *Q-Former* [5], a widely used architecture. It would also be helpful to include recent training-free compression approaches such as *VisionZip* [6] and *SparseVLM* [7], as the current baseline set appears somewhat outdated.

[1] VisionThink: Smart and Efficient Vision Language Model via Reinforcement Learning

[2] Are We Using the Right Benchmark: An Evaluation Framework for Visual Token Compression Methods

[3] Qwen2.5-VL Technical Report

[4] LLaVA-NeXT: Improved reasoning, OCR, and world knowledge

[5] BLIP-2: Bootstrapping Language-Image Pre-training with Frozen Image Encoders and Large Language Models

[6] VisionZip: Longer is Better but Not Necessary in Vision Language Models

[7] SparseVLM: Visual Token Sparsification for Efficient Vision-Language Model Inference

**Questions:**

See weaknesses

---

> ### Author Response · Authors · 2025-11-20
>
> Thanks for your constructive feedback.  We resolve your concerns as follows.
>
> - **Experiments on OCR-related tasks, such as OCRBench and ChartQA.**
>
>   Thanks for your helpful suggestion.
>
>   We evaluate our method on OCR-related tasks, including OCRBench and ChartQA benchmarks, and report the results as below.
>
>   | Method                       | OCRBench | ChartQA |
>   | :--------------------------- | :------: | :-----: |
>   | LLaVA-1.5-7B                 |   297    |  18.2   |
>   | STF -7B (ours, $k=2$, $E=1$) |   301    |  18.1   |
>
>   The results demonstrate that our method achieves comparable performance on resolution-sensitive benchmarks, OCRBench and ChartQA.
>
>   We supplement the above results in **Section 4.5 of revised paper**. We also supplement the discussion about the effect of token reduction [1, 2] in our revised paper.
>
>   [1] VisionThink: Smart and Efficient Vision Language Model via Reinforcement Learning
>   [2] Are We Using the Right Benchmark: An Evaluation Framework for Visual Token Compression Methods
>
> - **Direct extension on models like Qwen2.5-VL and LLaVA-NeXT.**
>
>   Since the training datasets and codes for both Qwen2.5-VL [3] and LLaVA-NeXT [4] are not published, we can't reproduce their results or directly apply our method on them in the same training settings as them.
>
>   The core contribution of our method is that our method can achieves comparable performance for LLaVA-1.5 when only 25% vision tokens are used to train and deploy the model.
>
>   [3] Qwen2.5-VL Technical Report
>
>   [4] LLaVA-NeXT: Improved reasoning, OCR, and world knowledge
>
> - **Comparison to other recent methods, including Q-Former, VisionZip and SparseVLM.**
>
>   We further compare our method with Q-Former [5], where 144 query tokens are used to replace 576 vision tokens of LLaVA-1.5-7B model (144 is 25% of 576) for fair comparison. Other hyper-parameters are consistent to the original LLaVA-1.5-7B model. The results are listed as follows.
>
>   |     Method     | GQA  | SQA  | VQAT | ROPE | MMB  | MMBCN | VQAv2 | VisWiz | Avg  |
>   | :------------: | :--: | :--: | :--: | :--: | :--: | :---: | :---: | :----: | :--: |
>   |    Q-Former    | 60.4 | 68.9 | 56.8 | 84.3 | 64.2 | 57.4  | 77.1  |  49.6  | 64.8 |
>   | **STF (ours)** | 61.9 | 70.5 | 57.4 | 86.0 | 66.2 | 58.9  | 78.1  |  51.1  | 66.3 |
>
>   The above results support that our method can significantly surpass Q-Former on 8 popular vision-language benchmarks.
>
>   We supplement the above results in **Section 4.4.3 of revised paper**.
>
>   We compare our method with with recent training-free compression approaches, such as VisionZip [6] and SparseVLM [7] as follows.
>
>   |     Method     |   GQA    |   SQA    |   VQAT   |   ROPE   |   MMB    |  VQAv2   |
>   | :------------: | :------: | :------: | :------: | :------: | :------: | :------: |
>   |   VisionZip    |   57.6   |   68.9   |   56.8   |   80.5   |   62.0   |   75.6   |
>   |   SparseVLM    |   56.0   |   67.1   |   54.9   |   62.0   |   60.0   |   73.8   |
>   | **STF (ours)** | **61.9** | **70.5** | **57.4** | **86.0** | **66.2** | **78.1** |
>
>   Experimental results demonstrate that our method achieves significantly better performance than training-free compression approaches.
>
>
>
>   [5] BLIP-2: Bootstrapping Language-Image Pre-training with Frozen Image Encoders and Large Language Models
>
>   [6] VisionZip: Longer is Better but Not Necessary in Vision Language Models
>
>   [7] SparseVLM: Visual Token Sparsification for Efficient Vision-Language Model Inference

---

### Official Review · Reviewer_fgaa · 2025-11-10

**Soundness:** 3
**Presentation:** 3
**Contribution:** 2
**Rating:** 4
**Confidence:** 5

**Summary:**

This paper studies the efficient large multi-modal models (LMMs). The author considers reducing the number of vision tokens fed into LLM and proposes the Spatial Token Fusion (STF) and Multi-Layer Token Fusion (MLTF) module to fuse adjacent vision tokens and capture multi-granularity visual features to obtain compact vision tokens. Extensive experiments have been conducted in standard vision-language datasets, where the proposed methods outperform the existing efficient LMM methods.

**Strengths:**

1. The proposed methods are reasonable and relevant for efficient LMMs. The motivation and design are clear and easy to understand.

2. The proposed method can effectively reduce the visual tokens while achieving comparable results to other methods on average. The ablation study also justifies the effectiveness of the proposed modules.

**Weaknesses:**

1. The proposed method's improvement is not significant enough. In Table 2, the MME metrics of the proposed method (STC) are significantly worse than those of FastV, but there is no elaboration on the performance loss. This makes it suspicious for the reviewer to be convinced that the proposed method can achieve the state-of-the-art results.

2. More vision-language tasks should be considered. The reduction of the vision token will inevitably lead to information loss on visual content, which may result in a larger performance drop on tasks that heavily rely on visual content, e.g., referral object detection. The author should consider adding a comparison of those tasks to demonstrate whether the proposed method remains effective.

3. Missing ablation study. The author mentioned that "directly fusing of all these features can not obviously improve the performance, but significantly increases the computation cost", while the claim has not been verified in the analysis section. The author should add this comparison to justify the claim.

**Questions:**

Please refer to the Weaknesses section.

---

> ### Author Response · Authors · 2025-11-20
>
> Thanks for your constructive feedback. We handle your concerns as follows.
>
> - **The analysis and performance comparison on MME between STF and FastV.**
>
>   Thanks for your comments.
>
>   The performance on MME benchmark of original models, LLaVA-1.5 (lora) and LLaVA-1.5 (tune), is 1476.9 and 1510.7. However, FastV (MME score: 1511.7) **outperforms** both of the above original models on **only MME benchmark**, but significantly worse than them on other 8 benchmarks.
>
>   In contrast, our STF **achieves comparable performance on all 9 benchmarks** as the original models. We argue that **the results of our STF is more in line with expected performance**. Moreover, **except MME, our STF significantly outperforms FastV on all other 8 popular benchmarks**.
>
>   Since FastV doesn't claim the details of finetuning dataset after pruning, we deem that higher score on MME comes from its finetuning dataset, which are different from the one for the training of LLaVA-1.5.
>
> - **More results on  vision-language tasks that heavily rely on visual content, e.g., referral object detection.**
>
>   Thanks for your helpful suggestion.
>
>   We agree that more fine-grained vision-language tasks, such as referral object detection (InstructDET [1]),  can well strengthen the soundness of our paper. However, since the training codes of InstructDET are not available, we can't reproduce the experimental results in a short rebuttal period. We have discussed referral object detection method, InstructDET in the **revised introduction section** of our paper and plan to supplement the results in our final revision.
>
>   For fine-grained vision-language tasks, we validate our method on OCR tasks, including OCRBench and ChartQA benchmarks, and report the results as below.
>
>   | Method                       | OCRBench | ChartQA |
>   | :--------------------------- | :------: | :-----: |
>   | LLaVA-1.5-7B                 |   297    |  18.2   |
>   | STF -7B (ours, $k=2$, $E=1$) |   301    |  18.1   |
>
>   The results demonstrate that our method achieves comparable performance on resolution-sensitive benchmarks, OCRBench and ChartQA .
>
>   We supplement the above results in **Section 4.5 of revised paper**.
>
>   [1] Dang et al. InstructDET: Diversifying Referring Object Detection with Generalized Instructions. ICLR 2024.
>
> - **Ablation study on feature fusion from all vision encoder layers.**
>
>   Thanks for your insightful comments.
>
>   We compare the performance of different numbers of selected blocks for fusion in MBTF module, including 8, 16 and 24 blocks (evenly sampled). The results are listed as follows.
>
>   |      Method      | GQA  | SQA  | VQAT | ROPE | MMB  | MMBCN | VQAv2 | VisWiz |   Avg    |
>   | :--------------: | :--: | :--: | :--: | :--: | :--: | :---: | :---: | :----: | :------: |
>   | Ours (8 blocks)  | 61.9 | 70.5 | 57.4 | 86.0 | 66.2 | 58.9  | 78.1  |  51.1  |   66.3   |
>   | Ours (16 blocks) | 62.1 | 70.4 | 57.8 | 86.2 | 66.4 | 59.0  | 77.9  |  51.0  | **66.4** |
>   | Ours (24 blocks) | 61.7 | 70.1 | 57.1 | 85.8 | 66.0 | 58.3  | 77.3  |  50.2  |   65.8   |
>
>   When we evenly sample features 16 blocks from vision encoder for MBTF, our method achieves the best performance. While sampling features from 24 blocks, the average performance drops. We speculate that the potential reason is that more input features cause significantly more parameters of MBTF, thus resulting in overfitting.
>
>   We supplement the above results in **Section 4.4.4 of revised paper**.

---

### Note · Authors · 2026-02-05

I have read and agree with the venue's withdrawal policy on behalf of myself and my co-authors.

---

### Meta-Review · Area_Chair_jEAp · 2026-01-05

**Summary:**

This paper initially received three negative scores (of 4, 4, 4) and one positive score of 6. After the rebuttal, Reviewer Gjz5 indicated that the authors offer perfunctory responses, which do not adequately address her/his concerns about (1) how to combine the text tokens to reduce the visual tokens and (2) the new comparison with TokenPacker & MinGemini. Meta reviewer agrees with the concerns from Reviewer Gjz5, and SOTA methods published recently, and generalization evaluation are not added for more comprehensive comparison (e.g., Applicability to Other VLMs commented by Reviewer k6si). The authors claim that ``lossless’’ token reduction, while the proposed STF fails to achieve lossless performance in multiple benchmark evaluations (e.g., Table 2). Moreover, the latency comparison without SOTA methods and fine-tuning cost of 18.6s per iteration with ~67% computation of LLaVA-1.5-7B model in the rebuttal does not well address the concerns from Reviewer kah7. The reduction of visual tokens benefits the prefilling efficiency, but not the decoding efficiency, especially on long-sequence generation. This leads to the practicality issues of the proposed method

According to the understanding of the Meta reviewer, the design of the proposed STF and MBTF module is not novel in the network architecture design. Moreover, the authors somewhat avoid the main concerns and fail to directly respond to them, which is indicated by Reviewer Gjz5. Based on the above analysis, it is not recommended to publish the current status.

**Reviewer Concerns:**

Part comparison with compression methods and ablation study commented by Reviewer k6si, Reviewer fgaa and Reviewer Gjz5.

**Reviewer Scores:**

All reviewers tend to maintain their scores with a high probability.

---

### Decision · Program_Chairs · 2026-01-26

Reject